

# MoviePop
## Interaktywna aplikacja ułatwiająca wybór filmów

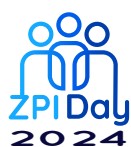

**Autorzy**: Izabela Kalenik · Sara Mac · Hanna Olechowska · Małgorzata Sielska · Magdalena Szeląg

**Opiekun:** Michał Szczepanik

**Streszczenie**

*MoviePop* to aplikacja mobilna, której celem jest uproszczenie procesu wyboru filmów do wspólnego oglądania. Rozwiązuje problem czasochłonnych dyskusji i trudności w dopasowaniu preferencji widzów. Dzięki intuicyjnemu mechanizmowi przesuwania plakatów w prawo (polubienie) lub w lewo (odrzucenie), użytkownicy mogą wyrażać swoje preferencje dotyczące filmów, a gdy kilka osób polubi ten sam tytuł, aplikacja może zarekomendować go do wspólnego seansu.

Funkcje, takie jak filtrowanie filmów po gatunkach czy sprawdzanie dostępności na platformach streamingowych, zapewniają wygodę i oszczędność czasu w szukaniu odpowiednich pozycji. Personalizacja preferencji zwiększa zaangażowanie użytkowników, czyniąc aplikację atrakcyjną dla szerokiej grupy odbiorców.

Aplikacja wypełnia lukę na polskim rynku, oferując unikalne rozwiązanie dla grup chcących wspólnie oglądać filmy. Projekt wyróżnia się na tle konkurencji, otwierając możliwości dla rozwoju społeczności lokalnych użytkowników i budowy przewagi rynkowej.

## 1 WPROWADZENIE

Długotrwałe dyskusje i trudności w podejmowaniu decyzji dotyczących wspólnego wyboru filmów są częstym wyzwaniem dla osób planujących seanse filmowe w grupie. Brak efektywnych narzędzi, które pozwalałyby na szybkie i wygodne wyrażenie preferencji oraz znalezienie pozycji odpowiadającej wszystkim oglądającym, dodatkowo utrudnia ten proces. Ponadto, indywidualni użytkownicy poszukują narzędzia, które pomoże im szybko odnaleźć filmy zgodne z ich gustem i dostępne na platformach streamingowych. Na polskim rynku brakuje aplikacji, która w pełni odpowiadałaby na te potrzeby, co stwarza szansę na wypełnienie tej luki.

### 1.1 Cele projektu

Celem projektu *MoviePop* było stworzenie aplikacji mobilnej, która uprości proces wspólnego wyboru filmów w grupie, jednocześnie oferując funkcjonalności przydatne dla indywidualnych użytkowników. W kontekście biznesowym projekt zakładał wypełnienie luki na rynku aplikacji rozrywkowych poprzez dostarczenie innowacyjnego i uniwersalnego narzędzia. Cele projektu obejmowały:

· **Uproszczenie wyrażania preferencji filmowych** – poprzez mechanizm przesuwania plakatów filmów w prawo (polubienie) lub w lewo (odrzucenie), użytkownicy mogą szybko i intuicyjnie wyrażać swoje preferencje na temat wyświetlanych pozycji.

· **Ułatwienie wyboru filmów w grupie** – dzięki funkcji *pokoju*, aplikacja umożliwia połączenie pomiędzy użytkownikami w celu znalezienia wspólnie polubionych filmów. W momencie dopasowania, aplikacja rekomenduje polubione pozycje.

· **Wsparcie indywidualnych użytkowników** – aplikacja umożliwia pojedynczym użytkownikom odkrywanie nowych filmów i szybki wybór pozycji zgodnych z ich preferencjami wraz z opcją losowego wyboru (shaker).

· **Personalizacja doświadczeń** – możliwość dostosowania preferencji, takich jak ulubione gatunki, aktorzy czy kryteria wyszukiwania, co zwiększa satysfakcję użytkowników.

· **Oszczędność czasu i wygoda** – dzięki intuicyjnemu interfejsowi, losowemu wyborowi filmów oraz możliwości sprawdzenia dostępności na platformach streamingowych, użytkownicy mogą szybciej podejmować decyzje.

## 1.2 Oczekiwane korzyści

Wdrożenie projektu przynosi istotne korzyści zarówno użytkownikom grupowym, jak i indywidualnym. Użytkownicy w grupach mogą zaoszczędzić czas i unikać konfliktów przy wyborze filmu, natomiast osoby korzystające z aplikacji solo mogą odkrywać nowe pozycje w prosty i wygodny sposób korzystając z dodatkowych funkcji dedykowanych dla użytkowników indywidualnych.

# 2 ROZWIĄZANIE

## 2.1 Istniejące rozwiązania

Na polskim rynku nie istnieją aplikacje oferujące funkcjonalności podobne do *MoviePop*, co pozwala naszej aplikacji wypełnić istniejącą lukę. Na rynkach zagranicznych można znaleźć aplikacje zbliżone pod względem celu, jednak często są one nieintuicyjne w obsłudze z uwagi na brak balansu między nadmiarem a niedoborem funkcji. Nasza aplikacja wyróżnia się przejrzystością, intuicyjnym interfejsem i prostotą obsługi, co zapewnia wygodę zarówno dla nowych, jak i doświadczonych użytkowników.

Jedną z kluczowych przewag konkurencyjnych *MoviePop* jest funkcja wspólnych pokojów, umożliwiająca użytkownikom identyfikację filmów, które zostały wspólnie polubione. Takie rozwiązanie jest unikalne i nie występuje w innych dostępnych aplikacjach.

Dodatkowym atutem jest dostępność w języku polskim i angielskim, co zwiększa potencjał aplikacji na lokalnym rynku i ułatwia jej adaptację przez szerszą grupę odbiorców.

## 2.2 Główne założenia projektowe

- **Platforma mobilna** - aplikacja została zaprojektowana przede wszystkim jako narzędzie mobilne, ponieważ jej funkcjonalności – takie jak szybki dostęp do filmów czy możliwość wspólnego korzystania z aplikacji w grupie – najlepiej sprawdzają się na urządzeniach przenośnych.

- **Technologia** - aplikacja została opracowana we frameworku *Flutter*, który dzięki wykorzystaniu języka programowania Dart umożliwia tworzenie aplikacji mobilnych działających zarówno na *Androidzie*, jak i *iOS*. Flutter to dynamicznie rozwijający się framework, który oferuje bogaty zestaw pakietów wspierających różne funkcjonalności.

- **Baza danych i uwierzytelnianie** - do przechowywania danych wykorzystano *Firebase*, co zapewnia skalowalność i stabilność aplikacji. Firebase umożliwia również łatwą implementację uwierzytelniania użytkowników za pomocą *Google SSO* i *Facebook SSO*.

- **Narzędzia i organizacja pracy zespołowej** - do implementacji projektu zespół korzystał z dwóch środowisk deweloperskich – *Android Studio* oraz *Visual Studio Code*, co umożliwiło elastyczną pracę nad kodem. Praca zespołowa była wspierana przez *Git* i *GitHub*, zautomatyzowane procesy CI/CD realizowano za pomocą *GitHub Actions*, a zarządzanie zadaniami i harmonogramem odbywało się w *Jira*. Dzięki temu projekt był realizowany w sposób uporządkowany i efektywny.

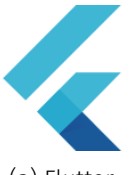 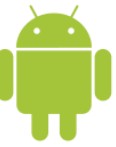 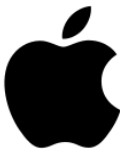 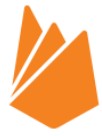 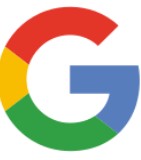 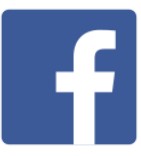

(a) Flutter   (b) Android   (c) Apple   (d) Firebase   (e) Google   (f) Facebook

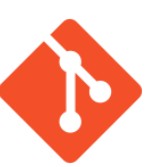 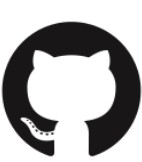 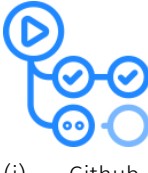 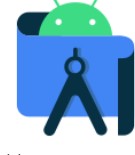 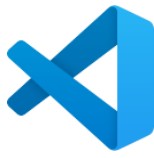 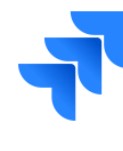

(g) Git   (h) Github   (i) Github-Actions   (j) Android Studio   (k) VS Code   (l) Jira

## 2.3 Trudności i ograniczenia

Podczas prac nad projektem zespół zmagał się z kilkoma wyzwaniami:

- **Nieaktualne pakiety** - dynamiczny rozwój Fluttera skutkuje częstymi aktualizacjami pakietów, co z jednej strony daje dostęp do nowoczesnych funkcji, ale z drugiej wymaga ciągłego researchu, aby uniknąć korzystania z przestarzałych rozwiązań. Niewielkie zasoby internetowe dotyczące nowych funkcjonalności utrudniały szybkie znajdowanie rozwiązań problemów.

- **Rozbudowana funkcjonalność** - choć aplikacja wyróżnia się prostotą interfejsu, wymagała jednoczesnego zaimplementowania szeregu funkcji współgrających ze sobą, co było wyzwaniem pod kątem organizacji i planowania pracy.

- **Ograniczenia czasowe** - projekt realizowany w ramach ZPI zakładał określone ramy czasowe, co skutkowało implementacją tylko koniecznych funkcji do prawidłowego działania aplikacji, pomijając te mniej użyteczne.

# 3  REZULTATY

## 3.1 Zaimplementowane funkcjonalności

W projekcie *MoviePop* udało się wdrożyć wszystkie kluczowe funkcjonalności, zapewniające prawidłowe działanie aplikacji. Główne elementy obejmują:

1. **Integracja z The Movie Database (TMDB) API**

   - Pobieranie danych o filmach, w tym tytułów, opisów, ocen, i dostępności na platformach streamingowych.
   - Dostępność polskich filmów i polskich platform streamingowych (np. Player), co znacząco zwiększa przyjazność aplikacji dla lokalnych użytkowników.
   - Możliwość wyświetlania treści w języku polskim.

2. **Przeglądanie i sortowanie filmów**

   - Funkcja **swipe** pozwalająca na wyrażanie preferencji filmowych – przesunięcie w prawo dodaje film do polubionych, a w lewo do odrzuconych.
   - Opcja **filtrowania preferencji** według gatunków, platform streamingowych, czy grup wiekowych.
   - Ekran szczegółowy filmu z informacjami, takimi jak opis, ocena czy dostępność na platformach streamingowych.

3. **Rekomendacja filmów na podstawie upodobań użytkownika**

   - Analiza preferencji filmowych – sugerowanie pozycji na podstawie ulubionych filmów użytkownika
   - Wykorzystanie **algorytmu rekomendacji** opartego na **"item-based collaborative filtering"** zaimplementowanego przez TMDB API, który:
     - oblicza **miarę podobieństwa** między filmami na podstawie tendencji ocen użytkowników (user ratings).
     - Na podstawie tej miary wybiera najbardziej podobne filmy i sugeruje je użytkownikowi.

4. **Funkcjonalność dla użytkowników indywidualnych**

   - Funkcja **shakera**, umożliwiająca losowy wybór filmu spośród ulubionych poprzez potrząśnięcie telefonem lub naciśnięcie przycisku.

5. **Funkcjonalność grupowa**

   - **Pokoje użytkowników** pozwalające na łączenie się osób i przeglądanie wspólnie polubionych filmów. Wspólne filmy są wyświetlane na osobnym ekranie, co ułatwia ich wybór na wspólny seans.

6. **Rejestracja i zarządzanie kontem użytkownika**

- Logowanie i rejestracja za pomocą e-maila, konta Google i konta Facebooka
- Edycja danych użytkownika (zdjęcia profilowego, nazwy użytkownika, kraju) oraz opcja resetowania hasła.
- Wybór języka aplikacji: polski lub angielski.

7. **Komunikacja z zespołem i zarządzanie kontem**

- Opcja raportowania problemów z aplikacją za pomocą e-maila.
- Możliwość usunięcia konta po kontakcie z obsługą przez e-mail.

8. **Powiadomienia**

- Aplikacja wysyła **powiadomienia pop-up** w kluczowych momentach, takich jak znalezienie wspólnie polubionych filmów czy zmiana hosta pokoju.

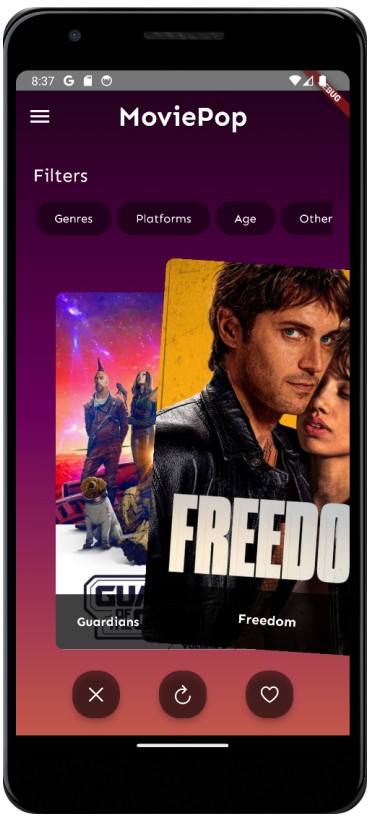

Rysunek 2: Ekran główny

## 3.2 Zrealizowane cele biznesowe i techniczne

- *Cel biznesowy:* Ułatwienie wyboru filmów zarówno dla grup użytkowników, jak i użytkowników indywidualnych.
- *Cel techniczny:* Opracowanie stabilnej, intuicyjnej aplikacji mobilnej opartej na frameworku Flutter, korzystającej z Firebase do uwierzytelniania oraz zarządzania danymi.
- Wdrożenie funkcji grupowych, takich jak wspólne pokoje, co wyróżnia aplikację na tle konkurencji.

## 3.3 Dane demonstrujące sukces projektu

- Pełna integracja z **TMDB API** umożliwiająca dostęp do danych o tysiącach filmów i ich filtrowanie według lokalnych preferencji.
- Implementacja płynnego działania kluczowych funkcji, takich jak **swipe**, **shaker** czy dołączenie do **wspólnego pokoju**.
- Dwujęzyczność aplikacji, zwiększająca jej dostępność dla szerokiego grona użytkowników.

- Powiadomienia popup znacząco poprawiające interakcję z użytkownikiem.

- Implementacja funkcjonalności **wyświetlania polubionych filmów**, umożliwiająca użytkownikom przeglądanie swoich polubionych tytułów.

### 3.3.1 Ankieta

Przeprowadzona przez nasz zespół ankieta pt. „*Ile czasu zajmuje Ci wybranie filmu do obejrzenia?*" pozwoliła lepiej zrozumieć potrzeby użytkowników i potwierdziła istnienie problemu, który nasza aplikacja *MoviePop* może rozwiązać. W badaniu wzięło udział 128 osób, głównie w wieku 18–25 lat (93% respondentów).

Wyniki ankiety wskazały, że wybór filmu indywidualnie zajmuje średnio od 5 do 20 minut, natomiast w przypadku grup decyzja ta często wydłuża się do 30 minut lub dłużej. To wyraźnie pokazuje, że proces wyboru filmu w grupie jest czasochłonny i mógłby zostać usprawniony.

Co więcej, aż 76,6% ankietowanych wyraziło zainteresowanie aplikacją, która ułatwia wspólne podejmowanie decyzji, a kolejne 13,3% nie wykluczało wypróbowania takiego rozwiązania. W kontekście użytkowników indywidualnych również zauważyliśmy potencjał – 52,3% osób wskazało, że byłoby zainteresowanych aplikacją wspierającą wybór filmów, a 33,6% uznało, że taka aplikacja mogłaby być dla nich użyteczna.

Wyniki te jednoznacznie wskazują, że młodzi ludzie dostrzegają problem w procesie wyboru filmu, szczególnie w grupie, i są otwarci na korzystanie z narzędzi, które mogą go usprawnić. *MoviePop* odpowiada na te potrzeby, oferując intuicyjne i szybkie rozwiązania, które nie tylko oszczędzają czas, ale także czynią cały proces przyjemniejszym.

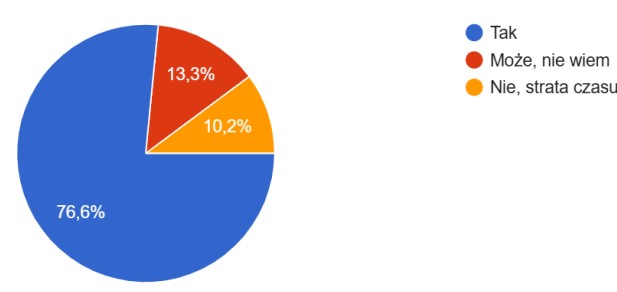

Rysunek 3: Pytanie z ankiety

*MoviePop* to projekt, który łączy technologię z praktycznymi potrzebami użytkowników, oferując innowacyjne rozwiązania, które upraszczają wybór filmów i zwiększają radość z procesu ich wyboru.

## 4  WNIOSKI

Projekt *MoviePop* zakończył się sukcesem, spełniając założone cele biznesowe i techniczne. Najważniejszym osiągnięciem jest stworzenie intuicyjnej i przejrzystej aplikacji mobilnej, która skutecznie odpowiada na potrzeby użytkowników indywidualnych i grupowych w procesie wyboru filmów. Zaimplementowana funkcja pokojów, umożliwiająca użytkownikom łączenie się i wyświetlanie wspólnie polubionych filmów, wyróżnia naszą aplikację na tle konkurencji, podkreślając jej innowacyjność i praktyczne zastosowanie.

Aplikacja oferuje szeroki zakres funkcjonalności, takich jak przeglądanie filmów, filtrowanie wyników, przesuwanie plakatów w celu wyrażenia preferencji, opcja shakera dla losowego wyboru filmów oraz powiadomienia informujące o wspólnie polubionych tytułach. Wyniki ankiety wykazały, że nasza aplikacja spotkała się z dużym zainteresowaniem potencjalnych użytkowników, co dodatkowo potwierdza trafność podjętego przez nas kierunku rozwoju.

## 4.1 Przyszłe kierunki rozwoju

Aby jeszcze bardziej rozwinąć i wzbogacić aplikację *MoviePop*, planujemy wprowadzenie nowych funkcji, takich jak:

- **Ekran obejrzanych filmów** – użytkownicy będą mogli śledzić historię obejrzanych tytułów, co ułatwi im unikanie powtórzeń i monitorowanie swoich preferencji.

- **Sesja na bieżąco** – dynamiczne wykrywanie wspólnie polubionych filmów w czasie rzeczywistym, co pozwoli na natychmiastowe wyświetlenie powiadomienia (popup) po dopasowaniu preferencji dwóch użytkowników.

- **Integracja z platformami streamingowymi** – możliwość przejścia bezpośrednio do serwisów takich jak Netflix czy Player za pomocą jednego kliknięcia w aplikacji, co znacząco zwiększy wygodę użytkowania.

- **Zwiastuny filmów** – użytkownicy będą mogli obejrzeć zwiastuny bezpośrednio na ekranie szczegółów filmu, co pomoże im lepiej ocenić wybrany tytuł.

- **Seriale** – rozszerzenie funkcjonalności aplikacji o obsługę seriali, umożliwiające użytkownikom przeglądanie, ocenianie i dodawanie do ulubionych swoich ulubionych tytułów z kategorii seriali.

Dalsze plany obejmują także rozszerzenie dostępnych języków oraz wdrożenie wersji webowej aplikacji, aby dotrzeć do szerszej grupy odbiorców. Dzięki tym ulepszeniom *MoviePop* ma szansę stać się jeszcze bardziej kompleksowym i wszechstronnym narzędziem dla miłośników filmów na całym świecie.

## LITERATURA

[1] API - The Movie Database. URL: `https://developer.themoviedb.org/docs/getting-started`.

[2] Dokumentacja Firebase. URL: `https://firebase.google.com/docs/flutter`.

[3] Dokumentacja Fluttera. URL: `https://docs.flutter.dev/`.

[4] Dokumentacja GitHub Actions. URL: `https://docs.github.com/en/actions`.

[5] Facebook - social technologies. URL: `https://developers.facebook.com/`.
