# OpenReview forum: "MoviePop - Interaktywna aplikacja ułatwiająca wybór filmów"
_pwr.edu.pl/Wrocław_University_of_Science_and_Technology/2024/ZPI_Day — Wrocław University of Science and Technology 2024 ZPI Day Submission_

### Official Review · Reviewer_pGy6 · 2024-12-05
**MoviePop - Interaktywna aplikacja ułatwiająca wybór filmów**

**Confidence:** 5
**Significance Of Results:** 5
**Overall Quality:** 5

**Compliance With Template:**

5: Very High Quality – The article contains all the required sections, which are written in a very detailed, clear, and error-free manner. The structure is professional and meets expectations, and the content adheres to the highest substantive and formal standards.

**Description Of Results:**

5: Very High Quality – The results are described in detail, clearly and comprehensively, supported by thorough evaluation, analysis, and convincing usage examples. The description meets the highest substantive standards.

**Feedback On Consistency:**

EN:
The project description of MoviePop is clear and logically organized. It starts with a well-defined problem analysis, identifying the challenges that groups and individuals face when selecting movies. The app’s features, such as the swipe mechanism and group rooms, are presented as direct solutions to these problems. The results section details the functional outcomes and user feedback, which align closely with the initial objectives. The conclusions effectively summarize how the app addresses the identified needs. Overall, there is strong consistency between the problem analysis, the presentation of results, and the conclusions, making the project easy to understand and follow.

The combination of a user-focused approach, strong technical foundation, and innovative features makes MoviePop a standout project with considerable potential for both user satisfaction and business success.

PL:
Opis projektu MoviePop jest przejrzysty i logicznie zorganizowany. Rozpoczyna się od dobrze zdefiniowanej analizy problemu, identyfikując wyzwania, przed którymi stoją grupy i jednostki podczas wybierania filmów. Funkcje aplikacji, takie jak mechanizm przesuwania i pokoje grupowe, są przedstawiane jako bezpośrednie rozwiązania tych problemów. Sekcja wyników zawiera szczegółowe informacje na temat wyników funkcjonalnych i opinii użytkowników, które są ściśle zgodne z początkowymi celami. Wnioski skutecznie podsumowują sposób, w jaki aplikacja odpowiada na zidentyfikowane potrzeby. W pracy występuje silna spójność między analizą problemu, prezentacją wyników i wnioskami, dzięki temu projekt jest łatwy do zrozumienia.

Połączenie podejścia skoncentrowanego na użytkowniku, solidnych podstaw technicznych i innowacyjnych funkcji sprawia, że MoviePop jest wyróżniającym się projektem o znacznym potencjale zarówno w zakresie zadowolenia użytkowników, jak i sukcesu biznesowego.

**Potential For Development:**

EN:
As a movie enthusiast, I recognize significant potential for further development of MoviePop. The article outlines several promising areas for enhancement:
- Expanded Content Support: Including TV series would attract a broader audience and increase user engagement.
- Streaming Integration: Direct connections with platforms like Netflix would streamline the viewing experience, making it more convenient for users to access recommended content.
- Gamification: Introducing rewards and leaderboards could boost user interaction and retention by adding a competitive and fun element to the app.
- Scalability and Global Reach: Expanding language support and developing a web-based version would make the app accessible to a wider, international audience.
- Community Features: Tools for organizing events and collaborating with local businesses could create additional value for users and open up new revenue streams.

By providing more detailed implementation plans for these future features, the project could enhance its roadmap and appeal to investors or partners. The app has substantial market potential, and these developments could position MoviePop as a leading solution in simplifying movie selection on a global scale.

PL:
Jako entuzjasta filmów dostrzegam znaczny potencjał dalszego rozwoju MoviePop. W artykule przedstawiono kilka obiecujących obszarów do ulepszenia i salszej pracy:
- Rozszerzona obsługa treści: Włączenie seriali telewizyjnych przyciągnęłoby szerszą publiczność i zwiększyło zaangażowanie użytkowników.
- Integracja streamingu: Bezpośrednie połączenie z platformami takimi jak Netflix usprawniłoby oglądanie filmów, ułatwiając użytkownikom dostęp do polecanych treści.
- Grywalizacja: Wprowadzenie nagród i rankingów mogłoby zwiększyć interakcję i retencję użytkowników, dodając do aplikacji element rywalizacji i zabawy.
- Skalowalność i globalny zasięg: Rozszerzenie obsługi języków i opracowanie wersji internetowej sprawiłoby, że aplikacja byłaby dostępna dla szerszej, międzynarodowej publiczności.
- Funkcje społecznościowe: Narzędzia do organizowania wydarzeń i współpracy z lokalnymi firmami mogłyby być dodatkową wartością dla użytkowników i otworzyć nowe możliwe źródła przychodów.

Przedstawiając bardziej szczegółowe plany wdrożenia tych przyszłych funkcji, projekt mógłby ulepszyć swoją "mapę drogową"(roadmap) i przyciągnąć inwestorów lub partnerów. Aplikacja ma znaczny potencjał rynkowy, a te zmiany mogą sprawić, że MoviePop stanie się wiodącym rozwiązaniem upraszczającym wybór filmów na skalę globalną.

**Project Nature Evaluation:**

EN:
The project presents key characteristics of engineering work. It applies advanced technical methods by using Flutter for cross-platform mobile development, ensuring the app is accessible on both Android and iOS devices. The integration of Firebase for secure data management and authentication demonstrates a focus on scalability and reliability. By incorporating the TMDB API, the app accesses extensive movie data, enhancing its utility. The use of recommendation algorithms showcases the application of sophisticated technological solutions to improve user experience. The project’s emphasis on user-centric design, combined with robust technical implementation, highlights its engineering maturity and practical utility.

PL:
Projekt wykazuje kluczowe cechy pracy inżynierskiej. Zastosowano w nim zaawansowane metody techniczne, wykorzystując Flutter do wieloplatformowego rozwoju mobilnego, zapewniając dostępność aplikacji zarówno na urządzeniach z systemem Android, jak i iOS. Integracja Firebase w celu bezpiecznego zarządzania danymi i uwierzytelniania świadczy o skupieniu się na skalowalności i niezawodności. Dzięki włączeniu API TMDB, aplikacja uzyskuje dostęp do obszernych danych filmowych, zwiększając jej użyteczność. Wykorzystanie algorytmów rekomendacji pokazuje zastosowanie zaawansowanych rozwiązań technologicznych w celu poprawy komfortu użytkowania. Nacisk projektu na projektowanie zorientowane na użytkownika, w połączeniu z solidną implementacją techniczną, podkreśla jego dojrzałość inżynieryjną i praktyczną użyteczność.

**Technical Language Precision:**

5: Very High Quality – The language is entirely appropriate for a technical report. All terms are used correctly and precisely, and the style is professional, clear, and coherent, without any errors or ambiguities.

---

### Official Review · Reviewer_ZPzu · 2024-12-06
**A review of a mobile application for movies selection.**

**Confidence:** 5
**Significance Of Results:** 4
**Overall Quality:** 3

**Compliance With Template:**

3: Average Quality – The article includes most of the required sections, but some may be incomplete, written in a general or unclear manner. The content is correct but requires further refinement.

**Description Of Results:**

4: High Quality – The results are described in detail and supported by usage examples or evaluations. The description is reliable but may lack full depth of analysis.

**Feedback On Consistency:**

The description of the work carried out in terms of the software engineering used is very sparse.

For example, there is no information about the system architecture used or the design patterns applied (if any are applied).

More technical details should be provided.

The description of the project is so general that it is difficult to do a fair review.

**Potential For Development:**

The project authors present a number of possible directions for further development work.

The potential development of the recommendation algorithms used is worth mentioning.

**Project Nature Evaluation:**

The project is an engineering work with possible commercialisation potential.

**Technical Language Precision:**

3: Average Quality – The language is mostly appropriate but may contain minor terminological or stylistic errors. Some statements might lack precision or require improvement for better readability.

---

### Official Review · Reviewer_yYeS · 2024-12-08
**Abstract of IT project done correctly.**

**Confidence:** 5
**Significance Of Results:** 5
**Overall Quality:** 5

**Compliance With Template:**

4: High Quality – The article contains all the required sections, which are well-written and substantively correct, although minor errors or shortcomings may be present. The overall structure is clear and coherent.

**Description Of Results:**

5: Very High Quality – The results are described in detail, clearly and comprehensively, supported by thorough evaluation, analysis, and convincing usage examples. The description meets the highest substantive standards.

**Feedback On Consistency:**

The analysis of the problem, presentation of results and conclusions are coherent and logical

**Potential For Development:**

The article indicates possibilities for further work or practical application of its results.

**Project Nature Evaluation:**

Both the level of usability, the applied technical methods and technological solutions have the characteristics of engineering work.

**Technical Language Precision:**

5: Very High Quality – The language is entirely appropriate for a technical report. All terms are used correctly and precisely, and the style is professional, clear, and coherent, without any errors or ambiguities.

---

### Decision · Program_Chairs · 2024-12-10

Accept (Oral)